# Hepatoblastoma Cancer Stem Cells Express PD-L1, Reveal Plasticity and Can Emerge upon Chemotherapy

**DOI:** 10.3390/cancers14235825

**Published:** 2022-11-25

**Authors:** Mieun Lee-Theilen, Delaine D. Fadini, Julia R. Hadhoud, Fleur van Dongen, Gabriela Kroll, Udo Rolle, Henning C. Fiegel

**Affiliations:** Department of Pediatric Surgery and Pediatric Urology, University Hospital, Goethe University Frankfurt, 60590 Frankfurt, Germany

**Keywords:** solid pediatric cancer, hepatoblastoma, cancer stem cells, CD34, CD90, csVimentin, OV-6, PD-L1 on CSCs, chemotherapy-induced CSCs, CSC plasticity

## Abstract

**Simple Summary:**

Cancer stem cells are thought to cause a poor response to chemotherapies. The aim of our study was to explore the still unknown biology of hepatoblastoma cancer stem cells that are essential for tumorigenesis. With our investigations, we aimed to gain more insight into cancer stem cell characteristics and to provide directions for new therapeutic approaches to treat refractory hepatoblastoma. We showed that hepatoblastoma cancer stem cells express PD-L1, a factor which helps tumors escape immune attacks. Furthermore, we detected cancer stem cell progeny evolving from non-cancer stem cells. Finally, we revealed that another subset of cancer stem cells is induced during chemotherapy. Our findings give a possible explanation why chemotherapies fail in certain hepatoblastoma cases and why new therapeutic approaches should consider the plasticity of hepatoblastoma cancer stem cells.

**Abstract:**

The biology of cancer stem cells (CSCs) of pediatric cancers, such as hepatoblastoma, is sparsely explored. This is mainly due to the very immature nature of these tumors, which complicates the distinction of CSCs from the other tumor cells. Previously, we identified a CSC population in hepatoblastoma cell lines expressing the CSC markers CD34 and CD90, cell surface Vimentin (csVimentin) and binding of OV-6. In this study, we detected the co-expression of the immune escape factor PD-L1 in the CSC population, whereas the other tumor cells remained negative. FACS data revealed that non-CSCs give rise to CSCs, reflecting plasticity of CSCs and non-CSCs in hepatoblastoma as seen in other tumors. When we treated cells with cisplatin and decitabine, a new CD34^+/low^OV-6^low^CD90^+^ population emerged that lacked csVimentin and PD-L1 expression. Expression analyses showed that this new CSC subset shared similar pluripotency and EMT features with the already-known CSCs. FACS results further revealed that this subset is also generated from non-CSCs. In conclusion, we showed that hepatoblastoma CSCs express PD-L1 and that the biology of hepatoblastoma CSCs is of a plastic nature. Chemotherapeutic treatment leads to another CSC subset, which is highly chemoresistant and could be responsible for a poor prognosis after postoperative chemotherapy.

## 1. Introduction

Hepatoblastoma is an embryonal tumor that is usually diagnosed before the age of 3 years [1,2]. The International Pediatric Liver Consensus Classification subdivides the tumors by histology in fetal, embryonal, cholangioblastic, macrotrabecular and small-cell undifferentiated (SCUD) hepatoblastoma [3]. These subtypes can be highly heterogeneous with closely intermixed histological components resembling different developmental stages [4,5,6,7]. Complete surgical resection is the mainstay of treatment. The anatomical tumor location, determined by the PRE- and POST-TEXT (Pre- and post-treatment extend of disease) classification, and its response to neoadjuvant chemotherapy defines the surgical resectability [3,8]. In most cases, total hepatectomy and liver transplantation can be avoided using chemotherapeutic pretreatment, which is also observed in the mature liver cancer type hepatocellular carcinoma [2,9]. Surgical resection is usually followed again by chemotherapy except in a few instances such as very low-risk hepatoblastomas [1]. The chemotherapy regimens are based on cisplatin but may also be extended using doxorubicin, vincristine, irinotecan and 5-fluorouracil [1]. Most hepatoblastomas are chemosensitive. Some hepatoblastomas, however, show a poor response to chemotherapy, such as the SCU subtype, and others develop chemoresistance after long-term therapy [2,10,11]. 

Cancer stem cells (CSC) can cause a poor response and resistance to chemotherapy. CSCs are considered to be the “beating heart” of a tumor and have central roles in tumor initiation, metastasis and relapse [12]. To this day, a great body of knowledge has been compiled on the CSCs of adult tumors. For example, research on CSCs in leukemia has shaped the commonly accepted roles of CSCs in tumorigenesis. Although, more and more studies are revealing differences in the CSCs of solid tumors [12,13]. 

Little is currently known about CSC markers of pediatric tumors. These tumors are mainly of embryonal nature, which makes it difficult to distinguish the CSCs from the other immature tumor cells [6,7,14,15]. Consequently, most classical CSC markers, which are common stem cell markers, are not automatically applicable for pediatric solid tumors such as hepatoblastoma. Despite these difficulties, we recently identified a CD34^+^OV-6^+^CD90^+^csVimentin^+^ population in hepatoblastoma cell lines, which showed CSC characteristics and susceptibility to the Hsp90 inhibitor 17-AAG [16]. In addition to the other well-known surface markers, Vimentin, a primarily intracellular protein was found to be expressed on the cell surface (csVimentin) of the CSC population. This ectopic expression was previously detected on hepatocellular carcinoma cells, indicating a key role in metastasis and providing a potential therapeutic target [17].

To date, the biology of hepatoblastoma CSCs is poorly investigated. However, recent studies have identified CSC-driving factors such as BEX1 and PIM3 [18,19]. RNA modifications, such as m6A, were also suggested to be involved in the hepatoblastoma CSC biology [20].

In recent years, a new form of tumor immune escape has been discovered. Cancer cells can shut down immune attacks by expressing the PD-L1 (programmed death-ligand 1) on their cell surface and interacting with the PD-1 (programmed cell death protein-1) receptor on tumor-infiltrating lymphocytes, such as the CD8+ cytotoxic T-cells. This leads to an inhibitory signal in the immune cells and suppression of the effector functions [21]. Previous studies observed an increased expression of PD-L1 on CSCs of various tumor types such as squamous cell carcinoma and breast and colon cancer [22,23,24]. 

Contemporary research has also shed new light on the source of CSCs. For the longest time, it was thought that only CSCs can give rise to new CSCs, which suggested that tumor cells belonged to a unidirectional hierarchy [13]. This model was undermined, however, with observations of melanoma and breast cancer cells. The conversion of these cancer cells was found to be bidirectional, not unidirectional, where non-CSCs de-differentiated and divided into new CSCs, suggesting plasticity of CSCs and non-CSCs [25,26]. 

In this study, we further explore the characteristics of our recently defined CD34^+^OV-6^+^CD90^+^csVimentin^+^ hepatoblastoma CSCs. We evaluated CSC PD-L1 expression, which enables tumor immune escape, and their behavior upon chemotherapeutic treatment. Additionally, we sought to gain insight into CSC regeneration and hierarchical organization.

## 2. Materials and Methods

### 2.1. Cell Culture

The HuH6 hepatoblastoma cell line was kindly provided by Prof. Kappler of the Pediatric Surgery Department, University Hospital Munich, Germany. The HepG2 hepatoblastoma cell line was kindly provided by Elsie Oppermann of the General Surgery Department, University Hospital Frankfurt am Main, Germany. Both cell lines were grown in Dulbecco’s Modified Eagle Medium + GlutamaxTM-I (Gibco, Carlsbad, CA, USA) at 37 °C, 5% CO_2_ and 90% humidity. The media was supplemented with 10% fetal bovine serum (Gibco, Carlsbad, CA, USA) and 2.5 mg/mL Gentamicin (Gibco, Carlsbad, CA, USA). Passaging of the cells was completed using incubation with Accutase (Sigma-Aldrich, St. Louis, MI, USA) for 10 min at RT. For the treatment with cisplatin (TEVA GmbH, Ulm, Germany), 1.5 × 10^6^ HuH6 cells or 4 × 10^6^ HepG2 cells were plated out the day before in 75 cm^2^ tissue culture flasks. The next day, the cells were treated with cisplatin at different concentrations for 72 h and subjected to further analyses. For the treatment with decitabine (Abcam, Berlin, Germany) and cisplatin in combination, the cells were plated out using the methods described for the treatment with cisplatin only. This treatment was also performed for 72 h. Magnetic separation of CD34-positive cells was performed according to the respective protocols provided by Miltenyi Biotec (Bergisch Gladbach, Germany).

### 2.2. Flow Cytometry

The HuH6 and HepG2 cells were stained for the expression of CD34 using a BV421-conjugated anti-CD34 antibody (clone 581, BD Horizon, Franklin Lakes, NJ, USA); the expression of CD90 using a FITC-conjugated anti-CD90 antibody (clone 5E10, BioLegend, Koblenz, Germany); the expression of Vimentin using a PE-conjugated anti-Vimentin antibody (clone D21H3, Cell Signaling Technology, Frankfurt, Germany) and the expression of PD-L1 using an APC-conjugated anti-PD-L1 antibody (clone 130021, R&D systems, Biotechne, Wiesbaden, Germany). The cells were also stained with an APC-conjugated OV-6 (Oval Cell marker) antibody (R&D Systems, Biotechne, Wiesbaden, Germany) or a PE-conjugated OV-6 antibody (Santa Cruz Biotechnology, Heidelberg, Germany). Flow cytometry analyses were performed using a BD FACSCanto II (BD Biosciences, San Jose, CA, USA). All data were analyzed using the Flowjo software v7.2.5 (BD Life Sciences, Ashland, OR, USA).

### 2.3. Fluorescence-Activated Cell Sorting

The HuH6 cells were simultaneously stained with a BV421-conjugated anti-CD34 antibody (clone 581, BD Horizon, Franklin Lakes, NJ, USA) and an APC-conjugated OV-6 antibody (R&D Systems, Biotechne, Wiesbaden, Germany). The CD34 and OV-6 double-negative cells were sorted in a BD Aria Fusion cytometer. The sorting gates were established using cells stained with isotype controls (Brilliant Violet 421 mouse IgG1 isotype control and APC mouse IgG1 isotype control, Biolegend, Koblenz, Germany). After the sorting, the collected CD34^−^OV-6^−^ and CD34^+^OV-6^+^ cells were re-analyzed for CD34 expression and OV-6 binding using the cytometer.

### 2.4. RNA Extraction and Transcription Analysis Using Real-Time PCR

As previously described [16], RNA was extracted from the cells using the Extractme Total RNA Kit (blirt S.A., Gdansk, Poland) according to the manufacturer’s protocol. cDNA was synthesized with the iScript cDNA Synthesis Kit (Bio-Rad Laboratories, Inc., Hercules, CA, USA). Real-time PCR analyses were performed using the iTaq Universal SYBR Green Supermix (Bio-Rad Laboratories, Inc., Hercules, CA, USA) with a Stratagene Mx3005P machine (Agilent, Santa Clara, CA, USA). qPCR data were calculated using the mean of two experimental replicates, and all qPCR experiments were repeated at least 3 times. Dissociation curves were generated to confirm the amplification of a single PCR product. All the quantifications were normalized to an endogenous ACTB control and calculated using the 2^−ΔΔCt^ method. All the primers used spanned introns and are listed in Appendix A [27,28,29,30]. Primers were designed using Primer3Plus or retrieved from publications.

### 2.5. Statistical Analysis

The results are presented in diagrams as means and standard deviations. A non-parametric Dunn’s multiple comparisons test was performed for comparisons between several groups. The *p*-values < 0.05 were considered statistically significant.

## 3. Results

### 3.1. CD34^+^OV-6^+^CD90^+^csVimentin^+^ Hepatoblastoma Cancer Stem Cells Are PD-L1 Positive

As previously shown [16], we identified a CSC population which co-expressed the cell surface markers CD34 and CD90 and cell surface Vimentin (csVimentin). Additionally, the CSCs were stained by the OV-6 antibody (i.e., the mouse monoclonal antibody OV-6), which recognizes a common epitope in Keratin 14 and Keratin 19 [31]. This population could be distinguished from the rest of the tumor cells, the non-CSCs, which were almost entirely CD34^−^OV-6^−^CD90^−^csVimentin^−^. 

Recent studies observed elevated levels of the immune escape factor PD-L1 on the CSCs of other tumor types. Therefore, we analyzed the hepatoblastoma cell lines, HuH6 and HepG2, for a potential PD-L1 expression using flow cytometry. We detected similar expression levels when compared to CD34, CD90, csVimentin and OV-6 binding. Simultaneous staining with α-CD34 and OV-6 antibodies revealed that PD-L1 was expressed on the same cells as CD34 and the OV-6 antigens and, hence, on the same cells as CD90 and csVimentin as shown in Figure 1 (the corresponding CD34/OV-6 dot blots are shown in Appendix A). In conclusion, the CSC subset could be extended using a fifth cell surface marker to become CD34^+^OV-6^+^CD90^+^csVimentin^+^PD-L1^+^.

### 3.2. CD34^−^OV-6^−^ Non-CSCs Give Rise to CD34^+^OV-6^+^CD90^+^csVimentin^+^PD-L1^+^ CSCs

To elucidate the source of the CSCs, we performed a fluorescence-activated cell sorting (FACS) to obtain CD34 and OV-6 double-negative non-CSCs. Therefore, we stained HuH6 cells with α-CD34 and OV-6 antibodies and sorted for CD34^−^OV-6^−^ cells, as shown in Figure 2A, Appendix A. We immediately plated the sorted and unsorted cells out in 6-well plates. After 48, 72 and 96 h of cultivation, we analyzed the cells using flow cytometry for CD34 expression and OV-6 binding. As CD34^+^OV-6^+^ cells co-express CD90, csVimentin and PD-L1, CD34 and OV-6 analyses were considered sufficient to detect CD34^+^OV-6^+^CD90^+^csVimentin^+^PD-L1^+^ cells. After 48 h, we observed CD34^+^OV-6^+^ CSCs in the CD34^−^OV-6^−^ sample with an average of 6.3% compared to the unsorted control sample with an average of 8.9%, as shown in Figure 2B,C. Interestingly, a certain percentage of cells in the CD34^−^OV-6^−^ sample showed various CD34^low^ and OV-6^low^ expression levels. After 72 h, the CD34^−^OV-6^−^ sample had even higher numbers of CD34^+^OV-6^+^ cells, which were almost as high as in the unsorted cells (8.1 versus 9.6%, respectively). A larger difference was apparent after 96 h, when the numbers in the unsorted sample rose to about 17.8%, whereas the numbers in the CD34^−^OV-6^−^ sample only rose to 10.5% on average.

### 3.3. A CD34^+/low^OV-6^low^ Population Emerges after Treatment with Cisplatin 

As CSCs are resistant to chemotherapeutic treatments, we treated HuH6 and HepG2 cells with cisplatin (1, 2.5, 5 and 7.5 µg/mL) for 72 h and analyzed the surviving cells for CD34 expression and OV-6 binding using flow cytometry (Figure 3A). The dot plots of both cell lines revealed two CD34^+^OV-6^+^ populations: a CD34^+/high^OV-6^high^ population (green line), which was the previously described CSCs, and a new CD34^+/low^OV-6^low^ (red line) population. The new CD34^+/low^OV-6^low^ population became more visible with higher cisplatin concentrations (5 µg/mL for HuH6 and 2.5 µg/mL for HepG2). The number of the CD34^+/low^OV-6^low^ cells strongly rose with higher cisplatin concentrations from 3.3% to 56% on average in HuH6 cells and from 1.1% to 46.6% in HepG2 cells. However, the percentage of the CD34^+/high^OV-6^high^ cells only increased within the initial cisplatin concentrations (1 and 2.5 µg/mL) from 4.7% to 16.7% in HuH6 cells and from 2.6% to 8.6% in HepG2 cells, which showed that the CD34^+/low^OV-6^low^ cells were responsible for the total increase of CD34^+^OV-6^+^ cells (Figure 3B). In conclusion, we observed a new CD34^+/low^OV-6^low^ subset which seems highly chemoresistant and can be distinguished from the previously identified CSC subset (CD34^+/high^OV-6^high^CD90^+^csVimentin^+^PD-L1^+^ cells).

### 3.4. Treatment with Cisplatin and Decitabine in Combination Results in Increased Numbers of CD34^+/low^OV-6^low^ Cells, Chemoresistant CD34^+/low^OV-6^low^ Cells Are CD90 Positive but csVimentin and PD-L1 Negative

Higher concentrations of cisplatin (5 and 7.5 µg/mL) unmask the CD34^+/low^OV-6^low^ population but result in only a few viable cells [16]. Therefore, we searched for another method to obtain a reasonable amount of these cells to investigate their profile. In previous attempts to specifically target CSCs, we treated the cells with decitabine and observed a slight yet promising inhibitory effect on our subset of CD34^+/high^OV-6^high^CD90^+^csVimentin^+^PD-L1^+^ cells. However, when we added this agent to a cisplatin treatment, the number of cells in the subset did not decrease, but, more interestingly, the number of the newly discovered CD34^+/low^OV-6^low^ CSC subset strongly increased compared to cisplatin mono-treated cells (Appendix A). Therefore, we decided that treatment with the hypomethylating agent decitabine would obtain a measurable amount of CD34^+/low^OV-6^low^ cells with the use of moderate cisplatin concentrations. We treated HuH6 with 100nM decitabine and 2 µg/mL cisplatin and HepG2 with 250 nM decitabine and 3 µg/mL cisplatin for 72 h and analyzed the surviving cells for CD34, CD90, csVimentin and PD-L1 expression and OV-6 binding using flow cytometry (Figure 4 and Appendix A). The results again showed two different CD34^+^OV-6^+^ cell populations with CD34^+/low^OV-6^low^ cells (Figure 4, red solid line) as the majority. Furthermore, we detected CD90 expression on the CD34^+/low^OV-6^low^ cells but did not detect csVimentin or PD-L1 expression, which differs from the CD34^+/high^OV-6^high^CD90^+^csVimentin^+^PD-L1^+^ CSC subset (green dotted line, also shown in Figure 1). We refer to this new subset as CD34^+/low^OV-6^low^CD90^+^ cells.

### 3.5. CD34^+/low^OV-6^low^CD90^+^ Cells Are Not More Pluripotent and Do Not Show Higher EMT Features Than CD34^+/high^OV-6^high^CD90^+^csVimentin^+^PD-L1^+^ Cells

We investigated the pluripotency and EMT profile of the CD34^+/low^OV-6^low^CD90^+^ cells and compared the results to the already examined CD34^+/high^OV-6^high^CD90^+^csVimentin^+^PD-L1^+^ CSCs. HuH6 cells were treated with 100 nM decitabine and 2 µg/mL cisplatin and HepG2 cells were treated with 250 nM decitabine and 3 µg/mL cisplatin for 72 h. Both sets of treated cells were then enriched for CD34-positive cells using MACS (magnetic-activated cell sorting). As shown in Figure 5A for HuH6 cells, the CD34 enriched fraction (CD34+, green bar) of the control cells had an average of 61.7% CD34^+^ cells compared to the unsorted control cells, which had an average of 20.3% CD34^+^ cells (white bar). The CD34+ fraction of the treated cells (DAC+Cis, red bar) had an average of 72.6% CD34^+^ cells compared to the unsorted treated cells (grey bar), which had an average of 36.7% CD34^+^ cells. For HepG2 cells, as shown in Figure 5B, the CD34+ fraction (green bar) of the control cells had an average of 65.2% CD34^+^ cells compared to the unsorted control cells (white bar) with 17.9%. The CD34+ fraction of the treated cells (red bar) had an average of 73.2% CD34^+^ cells compared to the unsorted treated sample (grey bar) with 38.2%.

As seen in the previous results, most of the CD34+ fraction of untreated control cells was CD34^+/high^OV-6^high^CD90^+^csVimentin^+^PD-L1^+^ for HuH6 (55.5% versus 8.5% of CD34^+/low^OV-6^low^CD90^+^) and HepG2 (80.4% versus 7.2% CD34^+/low^OV-6^low^CD90^+^) cells. On the other hand, most of the CD34+ fraction of treated cells was CD34^+/low^OV-6^low^CD90^+^ cells for HuH6 (55.4% versus 9.8% of CD34^+/high^OV-6^high^CD90^+^csVimentin^+^PD-L1^+^ cells) and HepG2 (58.2% versus 29.3% of CD34^+/high^OV-6^high^CD90^+^csVimentin^+^PD-L1^+^ cells) cells as shown in this representative experiment in Figure 5C,D and Appendix A. Based on these results, we assigned the CD34+ fraction of control cells to the CD34^+/high^OV-6^high^CD90^+^csVimentin^+^PD-L1^+^ cells and the CD34+ fraction of treated cells to the CD34^+/low^OV-6^low^CD90^+^ cells and compared them to each other in their profiles. 

We performed qPCR analyses to evaluate the expression of the pluripotency factors Oct4 and Nanog; the CSC markers CD34, KRT14 (one of the OV-6 antigens) and CD90; the EMT transcription factors SNAI1 and Twist1; the proto-oncogene c-myc; and either EpCAM, which is expressed in premature liver cells, or Albumin, which is expressed in mature hepatocytes. For HuH6 cells, the qPCR analyses (Figure 5E–M) revealed a significant increase in Oct4, Nanog, CD34 and KRT14 for the CD34+ sorted control fraction when compared to the unsorted control and/or the treated DAC+Cis cells. Furthermore, a significant increase in Oct4, CD34 and Twist1 was observed for the CD34+ sorted DAC+Cis fraction compared to the unsorted control and/or the DAC+Cis cells. However, none of the analyzed factors showed a significant difference between the CD34+ sorted fractions of the control and the DAC+Cis cells. For HepG2 cells, the qPCR results (Figure 5N–V) showed a significant increase in Oct4, Nanog, CD34, KRT14, CD90, SNAI1, Twist 1 and c-myc for the CD34+ sorted control fraction when compared to the unsorted control and/or DAC+Cis cells. The expression of Oct4, Nanog, CD34, KRT14, CD90, SNAI1, Twist1 and c-myc was significantly increased in the CD34+ sorted DAC+Cis fraction compared to the unsorted control and/or the DAC+Cis cells. Again, no significant difference for any of the factors was detected between the CD34+ sorted fractions except for Albumin, which was increased in the CD34+ sorted control fraction. In conclusion, the CD34^+/low^OV-6^low^CD90^+^ cells presented in the CD34+ sorted DAC+Cis sample seemed to have increased levels of pluripotency and EMT factors but did not differ significantly in their pluripotency or EMT levels compared to the CD34^+/high^OV-6^high^CD90^+^csVimentin^+^PD-L1^+^ cells. Thus, CD34^+/low^OV-6^low^CD90^+^ cells represent another subset of hepatoblastoma CSCs. 

### 3.6. CD34^−^OV-6^−^ Non-CSCs Give Rise to CD34^+/low^OV-6^low^CD90^+^ Cells when Treated with Cisplatin and Decitabine

We also explored the origin of the CD34^+/low^OV-6^low^CD90^+^ cells. For this analysis, we used FACS to sort for CD34^−^OV-6^−^ cells and then plated the cells out in 6-well plates as shown in Figure 2A and Appendix A. After 24 h, we treated the sorted CD34^−^OV-6^−^ and unsorted cells with 100 nM decitabine and 2 µg/mL cisplatin. Again, we analyzed the cells for CD34 expression and OV-6 binding after 24, 48 and 72 h of treatment using flow cytometry (Figure 6A and Appendix A). Forty-eight hours after sorting and 24 h after treatment, the numbers of CD34^+/high^OV6^high^CD90^+^csVimentin^+^PD-L1^+^ cells were almost identical between the unsorted and CD34^−^OV-6^−^ samples (9.2 versus 9.8% CD34^+/high^OV6^high^CD90^+^csVimentin^+^PD-L1^+^ cells, respectively) (Figure 6B). Throughout the time measurements, both samples showed comparable numbers of CD34^+/high^OV6^high^CD90^+^csVimentin^+^PD-L1^+^ cells, which slightly decreased to 8.1% in the unsorted sample versus 8.9% in the CD34^−^OV-6^−^ sample.

On the contrary, the number of the CD34^+/low^OV-6^low^CD90^+^ cells clearly increased in both samples from 6.6 to 19.5% in the unsorted sample and from 14.1 to 24.8% in the CD34^−^OV-6^−^ sample (Figure 6C). This result suggests that CD34^+/low^OV-6^low^CD90^+^ cells can derive from CD34^−^OV-6^−^ non-CSCs and indicates that this CSC subset is indeed more chemoresistant than the CD34^+/high^OV6^high^CD90^+^csVimentin^+^PD-L1^+^ CSC subset.

## 4. Discussion

It is well known that CSCs use several tools to escape attacks from the immune system. For example, CSCs can decrease the expression of molecules crucial for antigen presentation to T-cell receptors [32]. With PD-L1 expression, which is increased on the CSCs of certain cancers, the cells can suppress the effector functions of tumor-infiltrating lymphocytes [21,22,23,24]. We analyzed PD-L1 in hepatoblastoma cell lines and found that our previously investigated CSC population of CD34^+/high^OV-6^high^CD90^+^csVimentin^+^ cells express PD-L1, whereas non-CSC tumor cells were negative for PD-L1. A few studies have investigated PD-L1 in pediatric cancers, and most detected no or a weak PD-L1 expression on tumor specimens [33,34,35,36]. Only studies on pediatric gliomas revealed a higher detection rate of PD-L1 [37,38]. As shown by others in adult studies and in our present data, PD-L1 is preferentially expressed in CSCs, which represent a minority of the tumor cells. The general low to weak PD-L1 detection in pediatric cancer studies could be explained by the possibility that PD-L1 expression was analyzed in gross tumor specimens and not specifically in the CSC population. 

PD-L1 expression analyses of pediatric cancers can be used to assess the potential efficacy of anti-PD-L1 antibody therapy. This therapy has already shown successful treatment results for various adult tumors, but the efficacy is still under investigation for pediatric cancers. Monotherapy trials with only a few pediatric patients have shown rather disappointing results and, consequently, combinatorial strategies were suggested for treatment and are currently being investigated [39,40,41,42]. Previous PD-L1 expression analysis studies only included a few hepatoblastoma specimens. Therefore, further investigations with a focus on hepatoblastoma are needed to clarify the significance of the PD-L1/PD-1 pathway and its possible blockade in this pediatric tumor. Evaluating the PD-L1 expression in refractory SCU hepatoblastomas, which have very immature tumor cells, would provide particularly important information.

The origin and (self-)renewal of CSCs either follow a rigid hierarchical organization in a unidirectional fashion, as shown in glioblastoma, or reveal a certain plasticity in their hierarchy, as shown for colorectal CSCs [43,44,45]. Our FACS sorting analyses detected new CD34^+/high^OV-6^high^CD90^+^csVimentin^+^PD-L1^+^ cells which derived from non-CSCs. We conclude that hepatoblastoma CSCs reflect the plasticity model, which is bidirectional not unidirectional.

To investigate changes in CSC behavior, we treated the tumor cells with cisplatin and observed that a new CD34^+^OV-6^+^ population emerged on a lower expression level. It is possible that this subset may have existed before cisplatin treatment because CD90-enriched populations included this population under untreated conditions. Cisplatin treatment might only unmask the CD34^+^OV-6^+^ population through the vast cell death of non-CSCs, which would allow the percentage among the viable cell population to increase. It is interesting that this CD34^+/low^OV-6^low^ subset outnumbered the already known CSC subset at higher cisplatin concentrations, which indicates an increased chemoresistance. However, it could also be possible that the high numbers were due to an increased proliferation rate. Expression analyses of CD90, csVimentin and PD-L1 showed that the CD34^+/low^OV-6^low^ subset expresses CD90 but lacks csVimentin and PD-L1, which differs from the already investigated CD34^+/high^OV-6^high^CD90^+^csVimentin^+^PD-L1^+^ CSC population. Concerning the pluripotency and EMT status, we observed a comparable behavior of the CD34^+/low^OV-6^low^CD90^+^ population. Therefore, we propose that this cell subset is another CSC population in hepatoblastoma. 

As already mentioned, the combinatorial use of cisplatin with decitabine revealed unexpected observations. Previous studies using decitabine observed a reduction of bladder and breast cancer stem cells [46,47]. However, we found that the numbers of CD34^+/low^OV-6^low^CD90^+^ CSCs were tremendously increased when decitabine was added to cisplatin. These contrasting results could imply that hypomethylation mediated by this DNA-Methyltransferase-inhibitor does not lead to the predicted effects in pediatric cancers, such as hepatoblastoma. Further evaluation of these trends is needed.

The detection of CD34^+/low^OV-6^low^CD90^+^ CSCs using cisplatin treatment raised several questions about the source of this population. Were these CSCs the daughter cells of CD34^+/high^OV-6^high^CD90^+^csVimentin^+^PD-L1^+^ cells, which are the dominant CSC population under non-treated conditions? Our FACS results revealed that the CSCs were not daughter cells. When treated with cisplatin and decitabine, non-CSCs also gave rise to CD34^+/low^OV-6^low^CD90^+^ cells. This observation is supported by several studies showing that chemo- and radiotherapies induce an increased proliferation rate in pre-existing quiescent CSCs and also trigger non-CSCs to undergo de-differentiation and produce therapy-induced CSCs in breast and non-small cell lung cancer [12,48,49,50,51,52]. One can assume that this plasticity can be also seen in various other tumor types. 

Nonetheless, the possibility that CD34^+/low^OV-6^low^CD90^+^ cells can also derive from CD34^+/high^OV-6^high^CD90^+^csVimentin^+^PD-L1^+^ cells, and vice versa, is of great importance. Additionally, since chemotherapy applies exogenous stress to the cells, it would be interesting to determine if other types of stress, such as hypoxia or nutrient deprivation, can also cause the emergence of CD34^+/low^OV-6^low^CD90^+^ CSCs. 

Altogether, our in vitro findings show that the plasticity of hepatoblastoma CSCs may complicate new therapeutic approaches. Indeed, our results suggest that targeting intrinsic CSC features will not inhibit differentiated tumor cells from producing a new CSC progeny.

## 5. Conclusions

In summary, our findings suggest that CSC biology in hepatoblastoma is plastic and that several different CSC subsets may co-exist, as shown for other cancers [48,53,54,55]. Given the fact that CSCs were induced by cisplatin treatment, our data reinforce the idea that chemotherapy is a two-edged sword. On the one hand, the vast tumor mass will be reduced. On the other hand, residual cells could convert into new CSCs that are highly chemoresistant and give rise to differentiated daughter cells, which may lead to tumor regrowth. The newly identified CSC population of CD34^+/low^OV-6^low^CD90^+^ cells in this study lacks PD-L1 and csVimentin expression, thus eliminating the treatment target for anti-PD-L1 or anti-csVimentin antibodies [56]. This chemotherapy-induced CSC subpopulation may, therefore, be a treatment-resistant population of new hepatoblastoma CSCs [52,57,58]. Despite the difference in PD-L1 and csVimentin expression, both CSC subsets share CD34 and CD90 expression along with OV-6 binding [6,7]. This should be further analyzed as possible targets for hepatoblastoma CSCs. For example, target factors for chimeric antigen receptor (CAR) T-cell approaches is a new immune therapy and is currently under investigation for neuroblastoma in clinical trials [59,60,61,62]. 

## Figures and Tables

**Figure 1 cancers-14-05825-f001:**
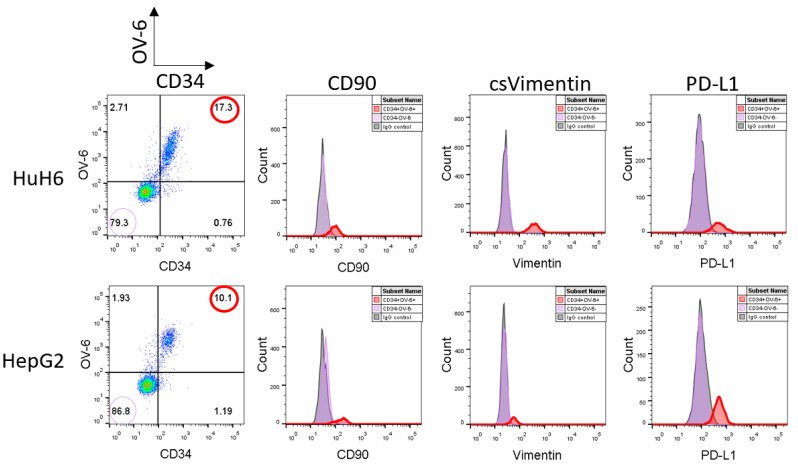
CD34^+^OV-6^+^CD90^+^csVimentin^+^ CSCs of hepatoblastoma cell lines express PD-L1. HuH6 and HepG2 cells were simultaneously analyzed for OV-6 binding and CD34 expression along with CD90, csVimentin and PD-L1 expression, respectively, using flow cytometry. The cells were first gated into CD34^+^OV-6^+^ (red line) and CD34^−^OV-6^−^ (violet line) cells and subsequently analyzed for CD90, csVimentin or PD-L1 expression. These are representative results of at least 5 experiments.

**Figure 2 cancers-14-05825-f002:**
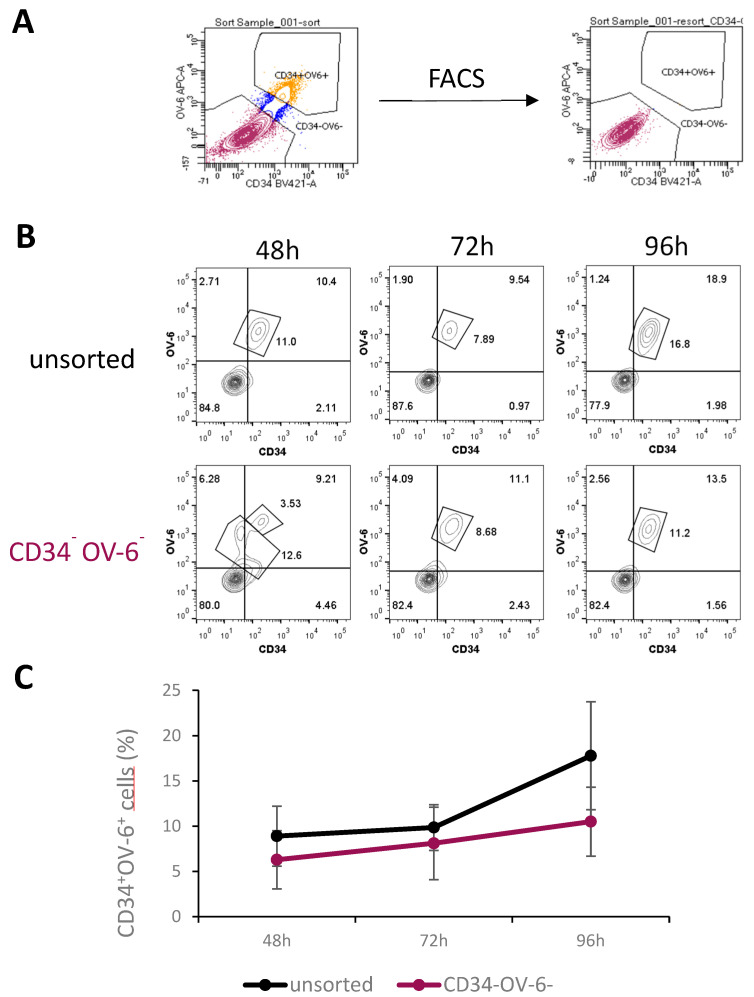
CD34^+^OV-6^+^CD90^+^csVimentin^+^PD-L1^+^ CSCs can originate from CD34^−^OV-6^−^ non-CSCs. (**A**) CD34^−^OV-6^−^ HuH6 cells were sorted by FACS and plated out in 6-well plates along with unsorted control cells. (**B**) After 48, 72 and 96 h, cells were measured for CD34 expression and OV-6 binding using flow cytometry. These are representative results of 5 experiments. (**C**) The values are the means, which are presented with error bars depicting the standard deviation from the mean.

**Figure 3 cancers-14-05825-f003:**
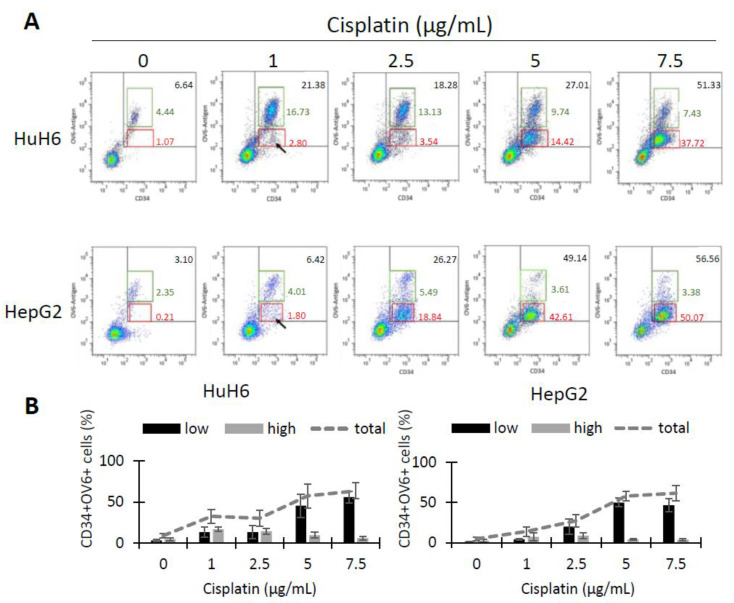
Cisplatin treatment reveals a second CD34^+^OV-6^+^ subpopulation with a lower expression. (**A**) HuH6 and HepG2 cells were treated with 0, 1, 2.5, 5 and 7.5 µg/mL cisplatin for 72 h. Surviving cells were analyzed for CD34 expression and OV-6 antibody binding by flow cytometry. The cells were gated in CD34^+/high^OV-6^high^ (green line) and CD34^+/low^OV-6^low^ (red line) cells. These are representative results of 7 experiments with HuH6 cells and 8 experiments with HepG2 cells. (**B**) The histograms depict the ratio of CD34^+/low^OV-6^low^ (low, black bars) and CD34^+/high^OV-6^high^ (high, grey bars) in each cisplatin concentration along with the total CD34^+^OV-6^+^ population (grey line). The columns represent the means with error bars depicting the standard deviation from the mean.

**Figure 4 cancers-14-05825-f004:**
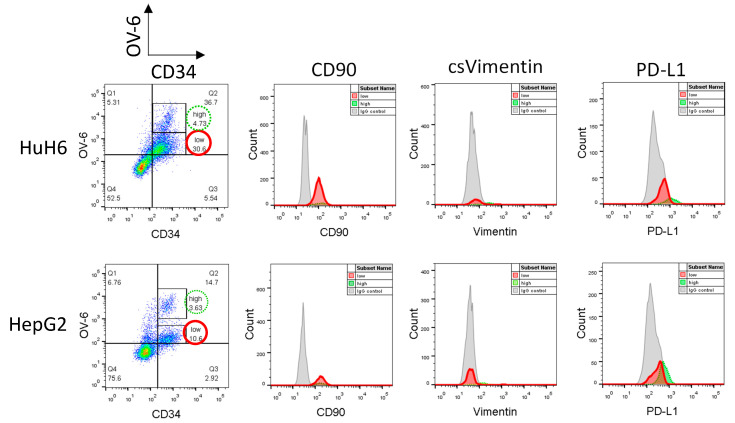
CD34^+/low^OV-6^low^ cells express CD90 but not csVimentin or PD-L1. HuH6 cells were treated with 100 nM decitabine and 2 µg/mL cisplatin and HepG2 cells with 250 nM decitabine and 3 µg/mL cisplatin for 72 h and measured for CD34, CD90, csVimentin and PD-L1 expression and for OV-6 binding using flow cytometry. The cells were first gated into CD34^+/high^OV-6^high^ (high, green line) and CD34^+/low^OV-6^low^ (low, red line) cells and subsequently analyzed for CD90, csVimentin or PD-L1 expression. These are representative results of at least 5 experiments.

**Figure 5 cancers-14-05825-f005:**
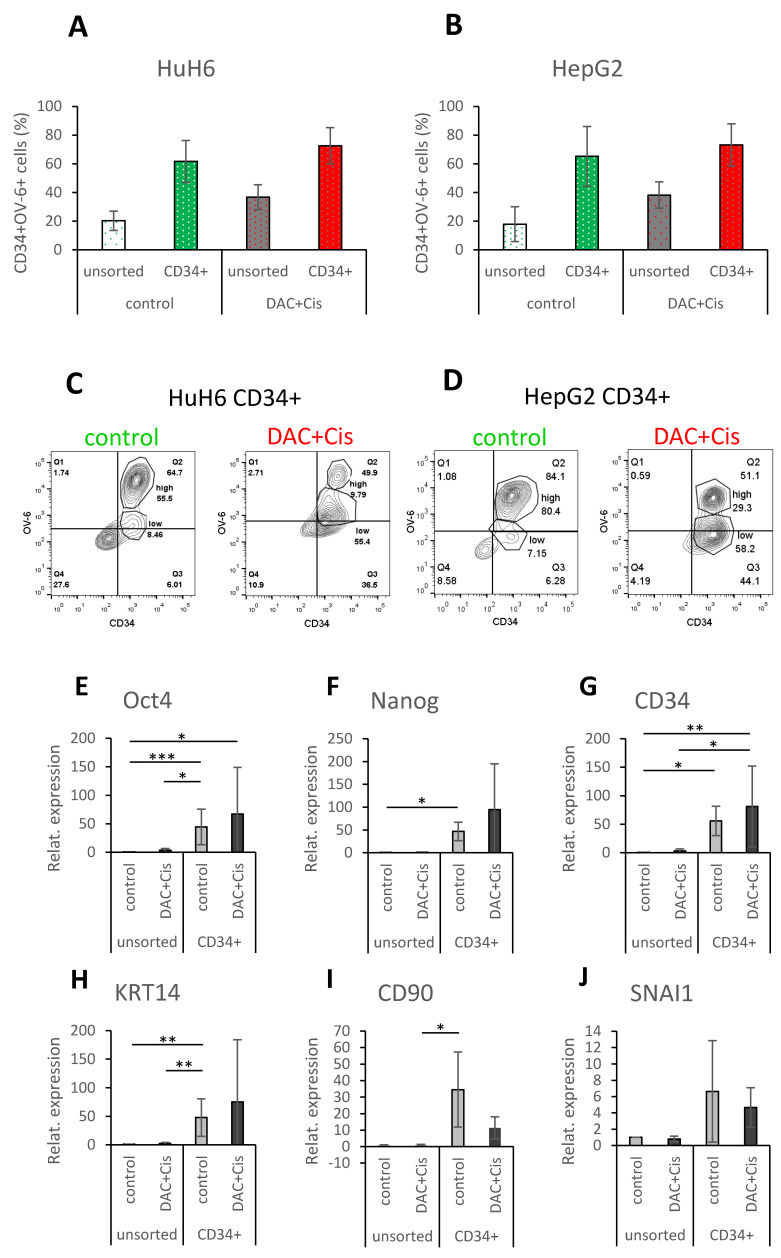
CD34^+/low^OV-6^low^CD90^+^ population reveals increased expression of pluripotency and EMT markers, but not compared to CD34^+/high^OV-6^high^CD90^+^csVimentin^+^PD-L1^+^ CSCs. HuH6 cells were treated with 100 nM decitabine and 2 µg/mL cisplatin, and HepG2 cells were treated with 250 nM decitabine and 3 µg/mL cisplatin. After 72 h, the cells were sorted for CD34 using the MACS technique. (**A**,**B**) Unsorted (white and grey) and CD34+ sorted (red and green) cells were analyzed for their CD34 expression and OV-6 binding using flow cytometry. The columns represent the means with error bars depicting the standard deviation from the mean (*n* = 9 for HuH6 and *n* = 8 for HepG2). (**C**,**D**) These are representative results of CD34 sorted fractions (CD34+) of the control and treated (DAC+Cis) cells. (**E**–**V**) Gene expression of Oct4, Nanog, CD34, KRT14, CD90, SNAI1, Twist1, c-myc and EpCAM or Albumin (**E**–**M** for HuH6 and **N**–**V** for HepG2) was analyzed using qPCR. The values of unsorted control cells were normalized to 1, and the values of the other samples were calculated accordingly. The columns represent the mean with error bars depicting the standard deviation from the mean. The experiment was repeated 7 times for HuH6 and 8 times for HepG2. A Dunn′s multiple comparisons test was performed to calculate the significance of the data (* *p* < 0.05, ** *p* = 0.001–0.01, *** *p* = 0.0001–0.001.

**Figure 6 cancers-14-05825-f006:**
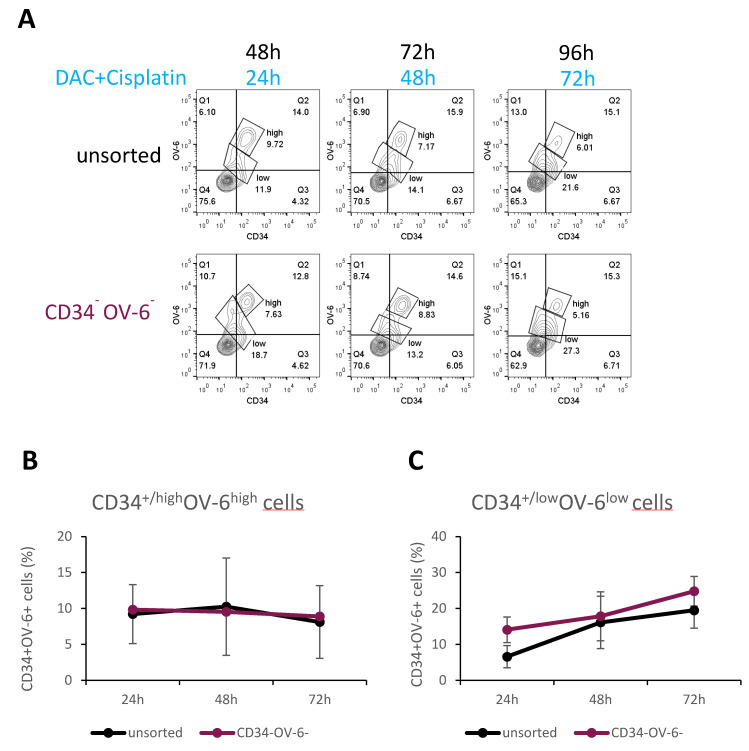
CD34^+/low^OV-6^low^CD90^+^ CSCs can derive from CD34^−^OV-6^−^ non-CSCs. (**A**) After 24 h of FACS sorting, CD34^−^OV-6^−^ and unsorted cells were treated with 100 nM decitabine and 2 µg/mL cisplatin for 24, 48 and 72 h and measured for CD34 expression and OV-6 binding using flow cytometry. These are representative results of 8 experiments. (**B**,**C**) The values are the means, which are presented with error bars depicting the standard deviation from the mean.

## Data Availability

The data presented in this study are available upon request from the corresponding author.

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
