# Peer review of "Hepatoblastoma Cancer Stem Cells Express PD-L1, Reveal Plasticity and Can Emerge upon Chemotherapy"

_cancers, 2022, doi:10.3390/cancers14235825_

Round 1

Reviewer 1 Report

I read with great interest your article. The Authors write a paper about the biology of cancer stem cells in hepatoblastoma. These cells express PD-L1, a factor which helps tumors to escape immune attacks. They reveal that another subset of cancer stem cells is induced upon chemotherapy. Our findings give a possible explanation why chemotherapies fail in certain hepatoblastoma cases and new therapy approaches should consider the plasticity of hepatoblastoma cancer stem cells.

    The message is clear, and your research is very original and I think it can give some input for a lot of future perspectives in cancer therapy and in neouadjuvant therapy.

By the way, I have some critical suggestions to do and I hope that the authors can take them into account, citing the literature proposed:   

1.      Can the author explain the role of vimentin in the metastatic process in the introduction? Is it important for Vascular invasion?

2.      Concerning hepatocellular carcinoma, the authors should discuss the role of neoadjuvant chemotherapy in HCC. The drug of choice used in the advanced-stage of HCC is sorafenib. Multiple nodules of HCC are an essential risk factor for recurrence, so neoadjuvant therapy could help in this sense. Transplant could be a solution but chemotherapy or surgery should be considered even if there is post-operative complication. (Mazzotta AD, et al Number of hepatocellular carcinoma nodules in patients listed for liver transplantation within alpha-fetoprotein score: a new prognostic risk factor. Transpl Int. 2021 May;34(5):954-963. doi: 10.1111/tri.13858. Epub 2021 Mar 15. PMID: 33660346.)

3.      Also for CAA there is some important future prospective, the authors should consider that this plasticity is also essential for cholangiocarcinoma, in fact, even in cholangiocarcinoma the chemotherapy treatment is sometimes indicated before surgery. Surgery in cholangiocarcinoma leads to complications, a study shows what is important to know in advance the risk factor via a nomogram (Golse N, et al.  Personalized Preoperative Nomograms Predicting Postoperative Risks after Resection of Perihilar Cholangiocarcinoma. World J Surg. 2020 Oct;44(10):3449-3460. doi: 10.1007/s00268-020-05618-8. PMID: 32474628.) But surely, the molecular expression in these tumour types must be integrated to better select patients. Can the authors add more in the discussion about that?

Reviewer 2 Report

In this manuscript, by utilizing FACS analysis, the authors have found that in hepatoblastoma cancer stem cells, a population also expresses the immune escape factor PD-L1. In addition, the authors have also found that the non-cancer stem cells could give rise to a cancer stem cell population. I think this finding is interesting and could give insights into other cancer stem cell studies. 

Reviewer 3 Report

This interesting manuscript presents two new results. First, the observation that hepatoblastoma CSC cells express PD-L1. Second the observation that cisplatin and decitabine-treated hepatoblastoma cells give rise to a CSC population that expressed a phenotyoe that differs, for some aspects, from the naturally occuring. The latter results is the more interesting of the two because it suggests that hepatoblastoma cells can give rise to two different CSC subpopulations. This would be clearly an important conclusion, but I am not totally convinced that the results allow such a drastic conclusion. Thus, one would like to know whether the chmoresistant CSC subpopulation (which I refer to as CSC2) can give rise to CSC1 after some culture period, and vice versa. The authors exlude the latter possibility but it would be interesting to see direct results with sorted CSC1 cells exposed to the chemotherapeutics. Moreover, chemotherapeutics represent an exogenous stress. There are other stresses that arise during tumor growth, such as mechanical stress, nutrient deprivation etc. It would be interesting to know if these stresses can also give rise to a CSC subpopulation similar to CSC2. Admittedly, performing experiments in order to address these points may take quite some time. Therefore, I would ask the authors to discuss these aspect. If they also want to perform some experiments, the better. 

Minor points. In the litreature there are some other articles from other gropus on hepatoblastoma CSCs. The authors should include the references and briefly discuss them. The manuscript is written in good English, but there are some typos that should be amended (e.g. a sentence should never start with And).
